# Natural variation in a glucuronosyltransferase modulates propionate sensitivity in a *C. elegans* propionic acidemia model

Huimin Na[1]☯, Stefan Zdraljevic[2]☯, Robyn E. Tanny[2], Albertha J. M. Walhout[1] *, Erik C. Andersen[2] *

**1** Program in Systems Biology and Program in Molecular Medicine, University of Massachusetts Medical School, Worcester, MA, United States of America, **2** Department of Molecular Biosciences, Northwestern University, Evanston, IL, United States of America

☯ These authors contributed equally to this work.
* marian.walhout@umassmed.edu (AJMW); erik.andersen@northwestern.edu (ECA)

**Data Availability Statement:** All relevant data are within the manuscript and its Supporting Information files.

## Abstract

Mutations in human metabolic genes can lead to rare diseases known as inborn errors of human metabolism. For instance, patients with loss-of-function mutations in either subunit of propionyl-CoA carboxylase suffer from propionic acidemia because they cannot catabolize propionate, leading to its harmful accumulation. Both the penetrance and expressivity of metabolic disorders can be modulated by genetic background. However, modifiers of these diseases are difficult to identify because of the lack of statistical power for rare diseases in human genetics. Here, we use a model of propionic acidemia in the nematode *Caenorhabditis elegans* to identify genetic modifiers of propionate sensitivity. Using genome-wide association (GWA) mapping across wild strains, we identify several genomic regions correlated with reduced propionate sensitivity. We find that natural variation in the putative glucuronosyltransferase GLCT-3, a homolog of human B3GAT, partly explains differences in propionate sensitivity in one of these genomic intervals. We demonstrate that loss-of-function alleles in *glct-3* render the animals less sensitive to propionate. Additionally, we find that *C. elegans* has an expansion of the *glct* gene family, suggesting that the number of members of this family could influence sensitivity to excess propionate. Our findings demonstrate that natural variation in genes that are not directly associated with propionate breakdown can modulate propionate sensitivity. Our study provides a framework for using *C. elegans* to characterize the contributions of genetic background in models of human inborn errors in metabolism.

## Introduction

Inborn errors of human metabolism are rare genetic diseases in which dietary nutrients or cellular metabolites cannot be broken down to generate energy, biomass, or remove toxic compounds. Most of these disorders are caused by loss-of-function mutations in genes encoding metabolic enzymes or metabolite transporters. Inborn errors of metabolism are often

**Funding:** AJMW and ECA, R01-DK115690, National Institutes of Heath, NIDDK The funders had no role in study design, data collection and analysis, decision to publish, or preparation of the manuscript.

**Competing interests:** The authors have declared that no competing interests exist.

considered monogenic disorders. However, the penetrance and expressivity of these diseases can vary [1]. Therefore, it has been proposed that such diseases should be viewed as more complex traits in which not only environmental factors such as diet, but also genetic background, affect the age of onset and severity of the disease [1]. If true, modifier genes could harbor variation in different genetic backgrounds and affect the penetrance and expressivity of metabolic disorders. However, because such diseases are rare, often with incidences of 1:50,000 or fewer, identifying modifier genes in human populations has been exceedingly difficult [1, 2].

Propionic and methylmalonic acidemia are inborn errors of metabolism in which the short-chain fatty acid propionate cannot be broken down [3]. Patients with propionic acidemia carry loss-of-function mutations in both copies of either one of two genes, PCCA or PCCB, which encode the two proteins comprising propionyl-CoA carboxylase that converts propionyl-CoA to D-methylmalonyl-CoA. Methylmalonic acidemia is a bit more complicated because it can be caused by mutations in either methylmalonyl-CoA racemase, methylmalonyl-CoA mutase, or in enzymes involved in the processing of vitamin B12, which is an essential cofactor for methylmalonyl-CoA mutase [3, 4]. Propionyl-CoA is generated in the natural breakdown of the branched-chain amino acids isoleucine and valine, as well as the catabolism of methionine, threonine, and odd-chain fatty acids. It can be inter-converted with propionate, which is generated by our gut microbiota during the digestion of plant fibers. Although propionate has been found to have beneficial functions [5, 6], it is toxic when it accumulates, as exemplified by patients with propionic acidemia [3]. Propionic acidemia is a rare disorder with a worldwide live birth incidence of 1:50,000 to 1:100,000. It is diagnosed in newborn screening by the detection of elevated levels of propionylcarnitine, 3-hydroxypropionate, and other aberrant metabolites [7].

The nematode *Caenorhabditis elegans* is a bacterivore found around the world [8–10]. In the laboratory, *C. elegans* can be fed different species and strains of bacteria [11, 12], but the vast majority of studies use the *Escherichia coli* strain OP50. However, *E. coli* OP50 cannot synthesize vitamin B12 and therefore cannot support the efficient breakdown of propionate by the canonical pathway, which depends on vitamin B12 [13, 14]. Previously, we found that *C. elegans* transcriptionally activates an alternative propionate breakdown pathway, or shunt, when flux through the canonical pathway is low due to genetic perturbations or because of low dietary vitamin B12 [15, 16]. This beta-oxidation pathway comprises five genes and generates acetyl-CoA [15](**Fig 1A**). *C. elegans* may have evolved a dedicated pathway for alternate propionate breakdown to be able to survive eating bacteria that do not synthesize vitamin B12. It only activates the expression of propionate shunt genes when propionate accumulation is persistent, via a specific regulatory circuit known as a type-1 feed-forward loop with AND-logic gate using the nuclear hormone receptors *nhr-10* and *nhr-68* [16]. In propionic acidemia patients, the buildup of propionate shunt metabolites indicates the presence of the propionate shunt. However, its activity is not sufficient to mitigate propionate toxicity likely because the enzymes functioning in other metabolic pathways are repurposed [15].

The vast majority of *C. elegans* studies rely on the laboratory-adapted strain named N2, which was isolated from Bristol, England [17]. Over the last twenty years, hundreds of *C. elegans* strains have been collected worldwide from natural habitats [10, 18–23]. *C. elegans* is a self-fertilizing hermaphrodite and, therefore, different wild strains can be easily maintained as fully isogenic strains. These different strains have been used to identify quantitative trait loci (QTL) that contribute to a variety of phenotypes, including anthelmintic and cancer chemotherapeutic resistance, and in several cases the precise genotypic variation that is causal to phenotypic variation has been determined [24–30]. Genomic information about the different strains is organized in the *C. elegans* Natural Diversity Resource (CeNDR), along with different tools for genome-wide association (GWA) mappings [31].

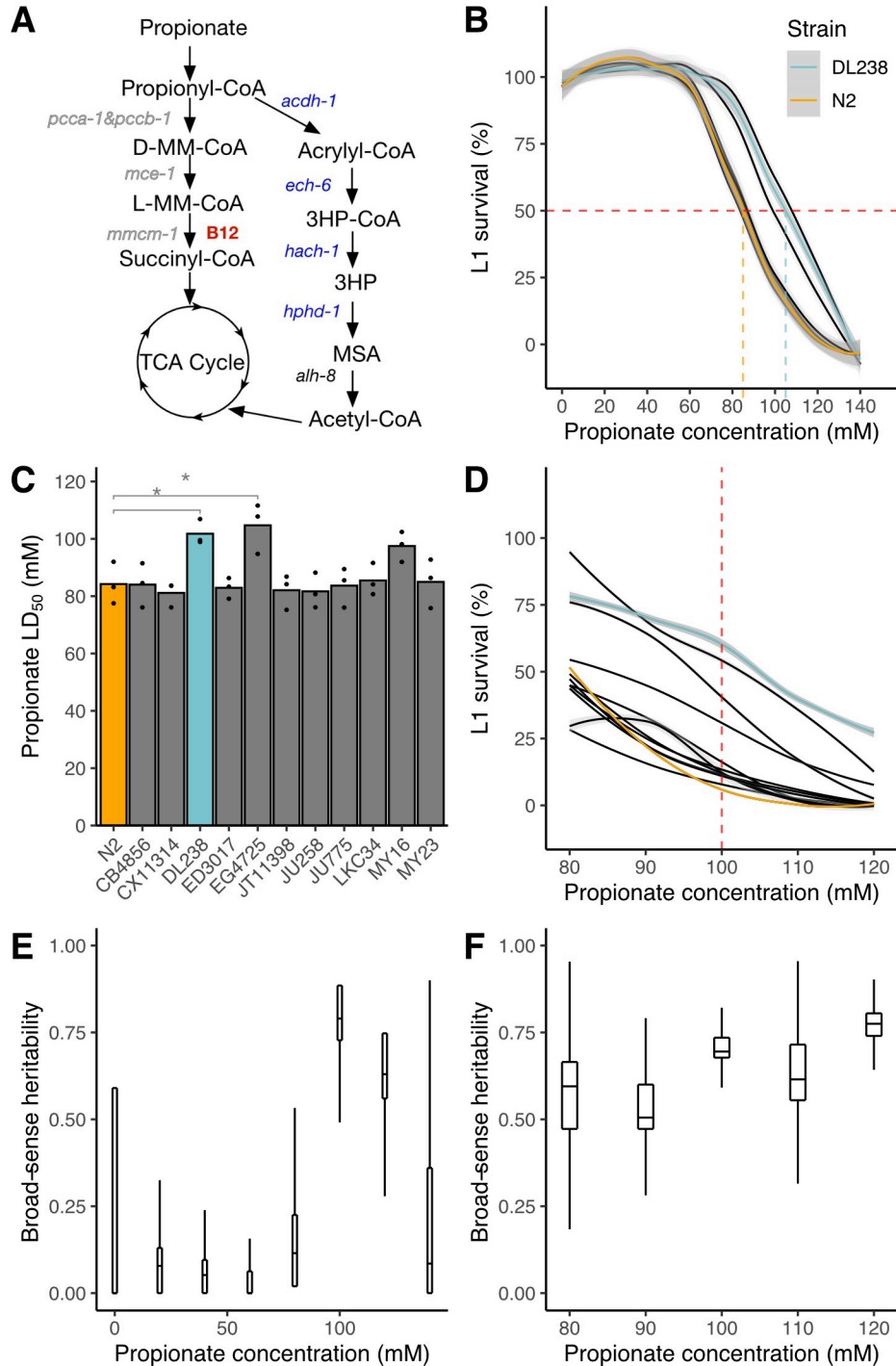

**Fig 1. Natural variation in propionate sensitivity in 12 genetically diverse *C. elegans* strains.** (A) Propionate breakdown pathways in *C. elegans*. MM–methylmalonyl; TCA–tricarboxylic acid; MSA–malonic semialdehyde; HP–hydroxypropionate. (B) Propionate dose-response curves (DRCs) for 12 genetically distinct wild *C. elegans* strains. The Loess-smoothed fits of three biological replicates, each comprising three technical replicates, is shown by solid colored lines, and the standard error of the fit is shown in gray. For reference, the DRCs are colored for the N2 (orange, propionate-sensitive) and DL238 (blue, propionate-resistant) strains. We note that the DRCs of nine strains overlap with the N2 DRC. The horizontal dashed red line indicates 50% L1 survival and the vertical colored lines represent the $LD_{50}$ concentration for N2 and DL238. (C) $LD_{50}$ values of L1 survival after propionate exposure for the 12 wild *C. elegans* strains. Three biological replicates each with three technical replicates were performed. (* indicates Student's t-Test $p < 0.05$. After adjusting for multiple testing, the p-value for the DL238—N2 comparison is 0.087 and the N2—

EG4725 comparison is 0.026, using Tukey's Honest Significant Difference method). (D) A new experiment of propionate DRCs of 12 wild *C. elegans* strains exposed to concentrations between 80 and 120 mM. The dashed red line indicates 100 mM propionate. For reference, the DRCs are colored for the N2 and DL238 strains as shown in (B). (E) Tukey boxplots of broad-sense heritability ($H^2$) estimates for the dose response in panel B. Each boxplot represents 12 $H^2$ estimates after subsampling three replicate measures. The interquartile ranges are cut off if they exceed the limits of the plotting window. (F) Tukey boxplots of broad-sense heritability ($H^2$) estimates for the dose response in panel (D). Each boxplot represents 12 $H^2$ estimates after subsampling three replicate measures.

Here, we used wild *C. elegans* strains to identify natural variation in loci that modify the resistance to exogenous propionate supplementation. Human propionic acidemia is caused by mutations in PCCA or PCCB. We have shown previously that *C. elegans* pcca-1 mutants provide a facile 'simple' model for this disease [14, 15]. It is impractical to generate *pcca-1* loss-of-function mutants in hundreds of wild *C. elegans* strains. However, we previously found that an *Escherichia coli* OP50 diet, which is low in vitamin B12, also mimics the metabolic conditions of propionic acidemia in *C. elegans* because flux through the canonical pathway is hampered when this vitamin is low [14, 15]. By supplementing propionic acid to the *C. elegans* diet, we can mimic propionic acidemia metabolic conditions [14, 15]. GWA mapping using 133 wild strains identified several independent genomic regions or QTL associated with propionate resistance. For one of these loci, we found the causal variant in *glct-3*, which encodes a predicted beta-1,3-glucuronosyltransferase, and is an ortholog of human B3GAT1, 2, and 3. A human homolog, B3GAT3, catalyzes the formation of the glycosaminoglycan-protein linkage by way of a glucuronyl transfer reaction in the final step of the biosynthesis of proteoglycans [32]. Glucuronosyltransferases also catalyze reactions between metabolites, specifically the addition of glucuronic acid to toxic metabolites such as drugs [33]. Interestingly, we found that loss-of-function mutations in *glct-3* confer resistance to propionate, indicating that it does not directly detoxify propionate. Our data show that quantitative toxicity phenotyping can be used to identify candidate modifier genes of traits associated with inborn errors in human metabolism.

## Results

### *C. elegans* wild strains differ in sensitivities to exogenous propionate

We previously developed a larval survival assay to test sensitivity of *C. elegans* to propionate supplementation [14–16]. In these assays, first larval stage (L1) animals are exposed to propionate and the proportion of animals that develop beyond that stage are quantified. Propionate dose-response curves (DRCs) showed that the laboratory-adapted strain N2 has an $LD_{50}$ of approximately 80 mM [14–16]. Supplementation of vitamin B12, which supports breakdown of propionate by the canonical pathway, confers resistance to propionate ($LD_{50}$ = 120 mM) and loss of the propionyl-CoA carboxylase ortholog (*pcca-1*) or the first gene of the propionate shunt (*acdh-1*) render the animals sensitive ($LD_{50}$ = 50 mM) [14, 15]. Because it is technically difficult to perform DRCs for all wild strains, we first asked whether 12 wild *C. elegans* strains, which represent high genetic diversity [19], exhibit differences in propionate sensitivity. To mimic metabolic conditions of human propionic acidemia, we fed the animals vitamin B12-depleted *E. coli* OP50 bacteria, which ensures that flux through the canonical propionate breakdown pathway was low [14, 15]. We performed three biological replicate experiments, each consisting of three technical replicates, and found that the 12 strains exhibited varying degrees of propionate sensitivity (**Fig 1B and 1C, S1 Table**). Nine of the strains had similar propionate sensitivities as the N2 strain with an $LD_{50}$ of approximately 85 mM. The other three strains were more resistant to propionate with an $LD_{50}$ of 100–110 mM. Next, we performed propionate assays at concentrations between 80 and 120 mM with 10 mM increments

and confirmed that most strains exhibited sensitivities similar to the N2 strain, but that the DL238 and EG4725 strains were significantly more resistant (**Fig 1C and 1D, S2 Table**). This result suggests that some wild strains have natural mechanisms to cope with high levels of propionate that are independent of vitamin B12 and the canonical propionate breakdown pathway.

To perform GWA mapping, we needed to test propionate sensitivity across a large set of wild *C. elegans* strains. Propionate sensitivity assays can be noisy, in part because of slight differences in experimental and environmental factors such as incubator and room temperature, propionate concentrations (which can change slightly due to evaporation and dilution), etc. To identify the dose with the highest reproducibility, we calculated broad-sense heritability ($H^2$) and found 100 mM propionate to be the best dose for the GWA mapping experiment ($H^2 = 0.74$) (**Fig 1E and 1F**). Additionally, we performed power analysis to determine the required number of replicate experiments prior to testing a large number of wild strains. We found that five independent experiments, each with four technical replicates, would give us 80% power to detect a 20% difference in propionate sensitivity (**S1 Fig**).

## Four genomic loci modify sensitivity to propionate across the *C. elegans* population

To identify the genetic basis of propionate response variation in *C. elegans*, we exposed 133 wild strains to 100 mM propionate and measured L1 survival (**S3 Table**). We tested the strains in three batches and included six strains in every batch to control for potential batch effects (**S2 Fig**). We observed a broad range of propionate sensitivities (**Fig 2A**). Using whole-genome sequence data and the L1 survival phenotype, we performed GWA mapping and identified four QTL that were above the Bonferroni-corrected significance threshold (**Fig 2B, S4 and S5 Tables**), one on chromosome II: (II:1880662–1993488); two on chromosome V: (V:3213649–4284434, V:19229887–19390858); and one on chromosome X: (X:9987812–10370303), coordinates are from WS245). To test the independence of these QTL, we calculated the pairwise linkage disequilibrium (LD) between each of the peak QTL markers (**S3 Fig**). We observed low levels of LD for the majority of QTL pairs, with the exception of the two QTL on chromosome V ($r^2 = 0.62$, peak markers—V: 3992679 and V: 19356375), suggesting that these two QTL might not be independent. Because multiple QTL were associated with propionate sensitivity, it was difficult to decide which QTL to characterize in more detail. Therefore, we used the sequence kernel association test (SKAT), which tests an association between the phenotype of interest and the cumulative variation on a gene-by-gene basis [34]. This approach identified two QTL, one that contains 14 genes significantly associated with propionate responses and overlaps with the QTL on the left arm of chromosome V (V:3213649–4284434) identified using the single-marker mapping approach (**Fig 2B**). The second QTL detected by SKAT was located on chromosome I and only overlaps with the single-marker mapping approach at a lower significance threshold (I:12110679–12591430) (**Fig 2C, S6 Table**). This additional support for the QTL on the left arm of chromosome V motivated us to investigate this genomic region further.

## Chromosome V near-isogenic lines do not recapitulate propionate resistance

To validate the effect of the QTL on the left arm of chromosome V, we constructed near-isogenic lines (NILs) in which the region associated with propionate resistance (V:3213649–4284434) was crossed from a resistant strain into the genome of a sensitive strain. To identify candidate parental strains for NIL construction, we focused on the 12 strains that were

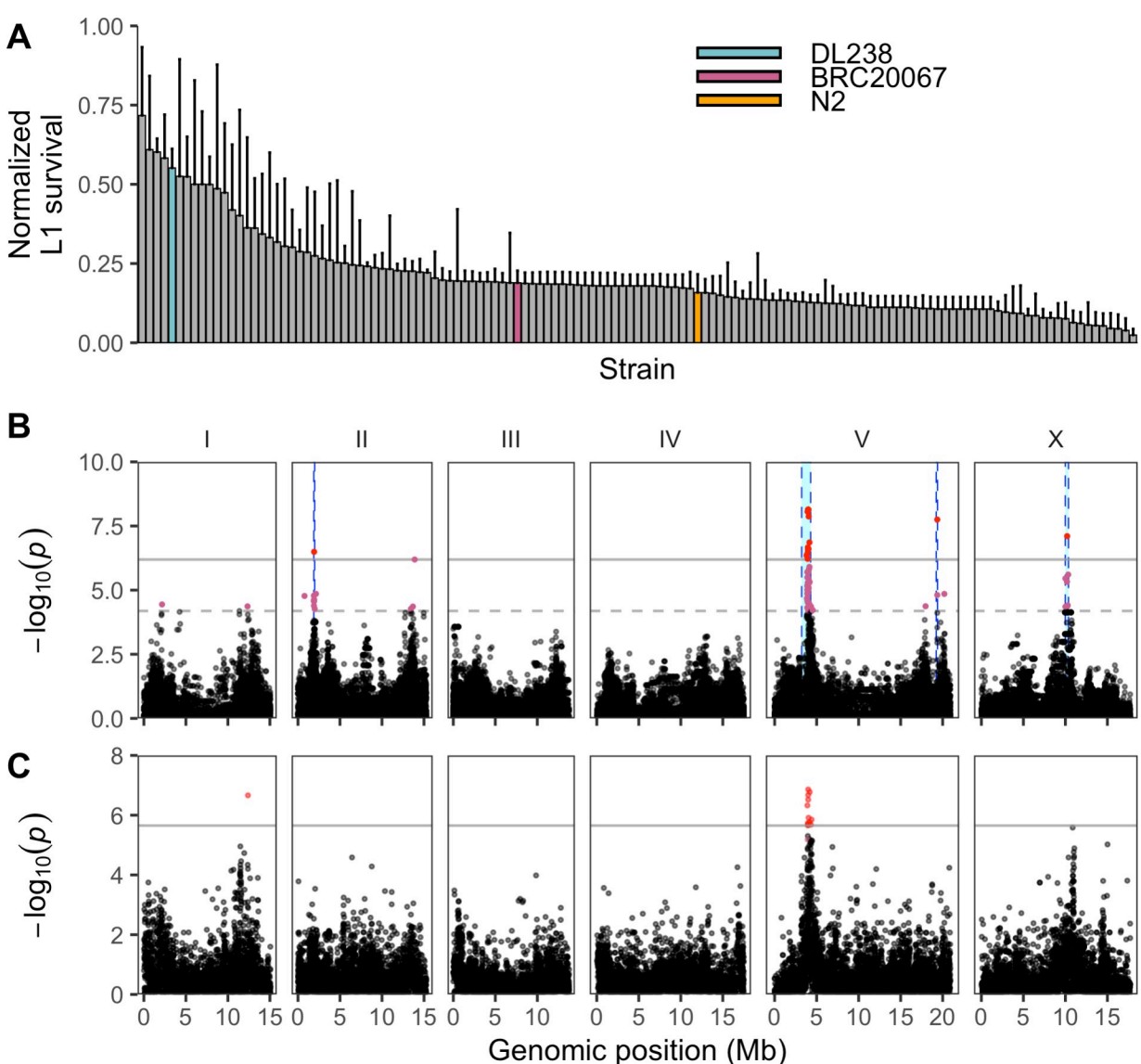

**Fig 2. Multiple QTL are associated with variable propionate sensitivities among *C. elegans* strains.** (A) Normalized L1 survival in the presence of 100 mM propionate for 133 wild *C. elegans* strains. L1 survival percentages were normalized by dividing each strain measurement by the maximum L1 survival percentage of all strains. Error bars show the standard deviation of replicate strain measurements. The reference strain N2 (orange) and the two strains discussed throughout this work DL238 (blue) and BRC20067 (pink) are colored. (B) Manhattan plot from marker-based GWA mapping for the normalized L1 survival percentage after propionate exposure. Each point represents an SNV that is present in at least 5% of the assayed wild population. The genomic position in Mb, separated by chromosome, is plotted on the x-axis and the *-log10(p)* for each SNV is plotted on the y-axis. SNVs are colored red if they pass the genome-wide Bonferroni-corrected significance (BF) threshold, which is denoted by the gray horizontal line. SNVs are colored pink if they pass the genome-wide Eigen-decomposition significance threshold, which is denoted by the dotted gray horizontal line. The genomic regions of interest surrounding the QTL that pass the BF threshold are indicated in cyan. (C) Manhattan plot from gene-based GWA mapping for L1 survival after propionate exposure. Each point represents a gene and is colored red if it passes the genome-wide BF threshold (gray line). Points are colored pink if they are significant after 1000 permutations. The genomic position in Mb, separated by chromosome, is plotted on the x-axis and the *-log10(p)* for each gene is plotted on the y-axis.

phenotyped in the dose-response experiment (**Fig 1C**). Of these 12 strains, two were significantly resistant to propionate and ten were sensitive. Next, we verified that the propionate-resistant strains had the alternative genotype at the peak QTL marker identified using the single-marker mapping method and were compatible with propionate sensitive strains at the *peel-1 zeel-1* [35] and *sup-35 pha-1* [36] incompatibility loci. Using these criteria, we identified

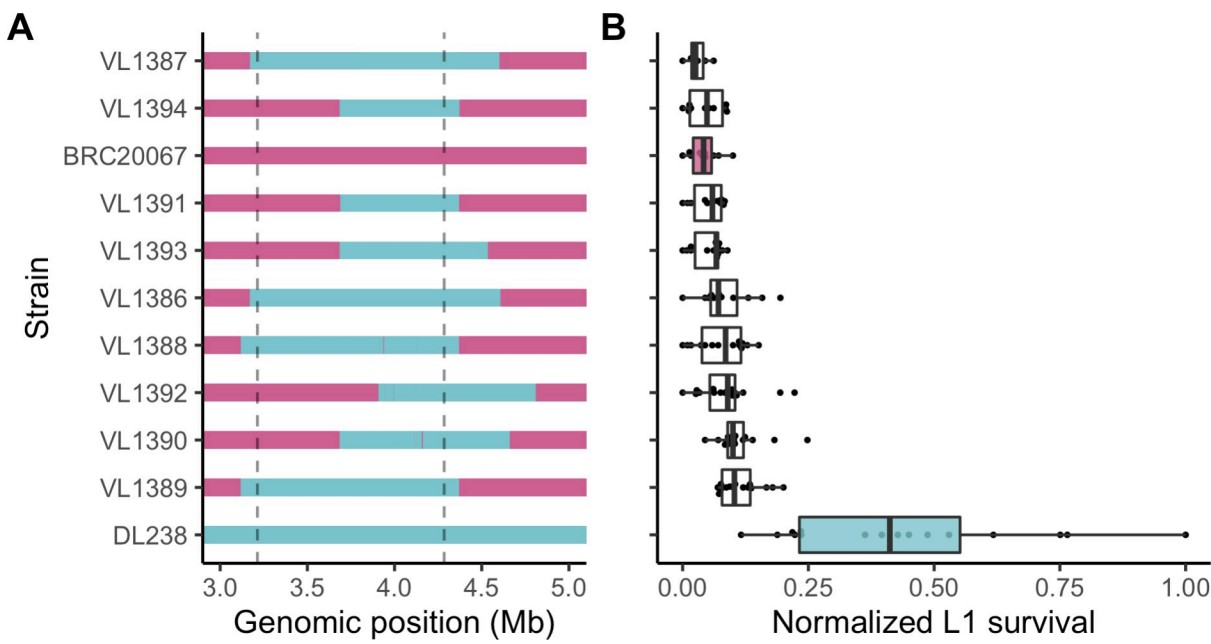

**Fig 3. Chromosome V near-isogenic lines do not recapitulate the chrV right QTL effect.** (A) Chromosome V genotypes of near-isogenic lines (NILs) generated between BRC20067 (pink) and DL238 (blue). The dotted lines denote the QTL region of interest from the marker-based GWA mapping. (B) Tukey box plots of normalized L1 survival after exposure to 100 mM propionate phenotypes of each NIL and parental strain. Each dot represents a replicate L1 survival measurement.

DL238 (propionate-resistant) and BRC20067 (propionate-sensitive) as suitable parental strains for NIL construction. We constructed nine NILs that contained the DL238 genomic region surrounding the chromosome V QTL introgressed into the BRC20067 genetic background (**Fig 3A**, **S7 Table**). When we exposed these NILs to propionate, we observed that the DL238 introgressed regions that correspond to the chromosome V QTL region of interest did not confer propionate resistance (**Fig 3B**, **S8 Table**). Because the genomic region spanned by these NILs is larger than the QTL region of interest, these results suggested that the chromosome V QTL we identified might have been the result of a spurious association with the QTL on the right of chromosome V. The LD between these two loci supports this hypothesis. Alternatively, because this genomic region has thousands of variants and we chose DL238 and BRC20067 based on their phenotype at the peak QTL marker, these two strains might not differ at the causal locus found in this QTL. Because the chromosome V QTL might have a complex relationship, we focused on the QTL identified on the right of chromosome I where the gene-based mapping overlapped the marker-based GWA mapping at the lower significance threshold (**Fig 2B and 2C**).

## Variation in the *glct-3* gene confers propionate resistance

Using a gene-based GWA mapping strategy, we found that variation in the gene *glct-3*, which is located on the right arm of chromosome I, is associated with propionate sensitivity among the wild isolates (**Fig 2C**). Additionally, the most correlated marker from the marker-based GWA mapping is in close proximity to *glct-3* (**Fig 4A**). Within the *glct-3* gene, we observed eight distinct combinations of alleles (haplotypes) among the phenotyped wild strains (**S4 Fig**). Five of these distinct haplotypes all include the same stop-gained variant at amino acid position 16 (Gly16*), along with other variants (Gly17Arg, Ser50Ala, Ser111Thr, Tyr231Cys in the QX1793 strain; Gly17Arg Ser50Ala in the CX11276, DL238, ED3046, ED3049, and

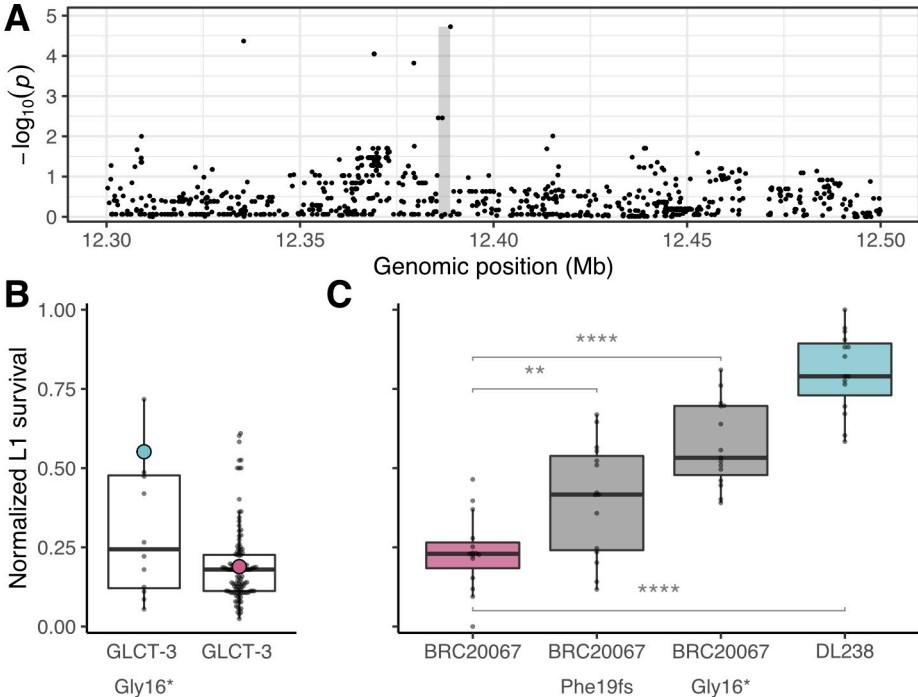

**Fig 4. Variation in *glct-3* underlies differential propionate sensitivity in *C. elegans*.** (A) Manhattan plot showing the strength of correlation between variants surrounding the *glct-3* gene identified by gene-based GWA mapping of the normalized L1 survival after propionate exposure phenotype. The gray shaded rectangle represents the *glct-3* gene (chrI:12385765–12388791). (B) Tukey box plots of *C. elegans* wild isolate's normalized L1 survival after exposure to 100 mM propionate. Each dot represents the mean of 20 replicate measures for each strain. Strains are separated by the presence of a stop-gained variant at amino acid position 16 of GLCT-3. DL238 (blue) and BRC20067 (pink) are highlighted for reference. (C) Tukey box plots of normalized L1 survival of each CRISPR-edited and parental strain after exposure to 100 mM propionate are shown. Each dot represents a replicate L1 survival measurement. (The effect of strain on phenotype data is significant from analysis of variance [F(3,56) = 45.081, $p < 0.05E\text{-}10$]. For individual strain comparisons: ** represents [F(1,28) = 9.28, $p = 0.005$] and **** for the BRC20067 –BRC20067 (Gly16*) comparison [F(1,28) = 55.25992, $p < 0.05E\text{-}6$]) And for the BRC20067 –DL238 comparison [F(1,28) = 165.7699, $p < 0.05E\text{-}6$]).

NIC252 strains; Leu184Phe in the ECA36 strain; Ile46Thr in the QX1792 strain, and only Gly16* in the MY23 and QX1791 strains). Strains with the Gly16* variant are 20% more resistant to propionate treatment than strains with no variation in *glct-3* (**Fig 4B**). This genomic region explains 13.1% of the total genetic variation in response to exogenous propionate, indicating that, although other loci contribute to this trait, this gene is a major contributor to natural differences in resistance to propionate.

To test whether variation in the *glct-3* is causal for the difference in propionate sensitivity, we generated two independent *glct-3* alleles in the propionate sensitive BRC20067 strain using CRISPR-Cas9 genome editing [37]. The *ww62* allele has a one base pair deletion at position 57 in the first exon that causes a frameshift in the reading frame leading to an early stop codon, and the *ww63* allele contains the same Gly16* variant found in DL238. In line with previous experiments, we found that DL238 was more resistant to propionate treatment than BRC20067 (Cohen's F = 2.433). The strains harboring the *ww62* and *ww63* alleles recapitulate 23.7% and 57.7% of the difference in propionate sensitivity between DL238 and BRC20067 as measured by Cohen's F, respectively (**Fig 4C**, **S9 Table**), demonstrating that the loss of *glct-3* function confers resistance to propionate. The QTL effect size that corresponds to a loss of *glct-3* function is approximately 10% of the total parental difference observed in this

experiment. We note that this estimate is similar to the fraction of the total heritability explained by the GLCT-3 Gly16* allele (0.11 of broad-sense heritability). We hypothesize that the discrepancy between the QTL effect in the GWA mapping experiment and the genome-edited strains might be caused by limited epistatic interactions because we introduced the allele in a single genetic background. In agreement with this hypothesis, when we only consider the additive component of the heritability ($h^2$) in the GWA experiment, we find that the GLCT-3 Gly16* allele accounts for approximately 40% of this additive trait variance. Finally, we note that the experimental setup for this follow-up experiment was much simpler than the GWA mapping experiment, which likely contributes to the discrepancy between explanatory power of the GLCT-3 Gly16* between the GWA mapping and genome-editing follow-up experiments.

To understand how variation in *glct-3* might have arisen in the *C. elegans* species, we investigated the frequency of this allele across the population and the strains that harbor strongly deleterious variants. More than 330 wild *C. elegans* strains are currently available in CeNDR [31], and 42 of these strains contain the Gly16* variant in *glct-3*. The majority of strains that contain the Gly16* variant (33/42) were isolated on the Hawaiian islands (S5 Fig), which are known to harbor the most genetically divergent *C. elegans* individuals [10]. An additional three strains have variants that are predicted to cause a loss of *glct-3* function (ECA733, JU1395, and ECA723). In agreement with the geographic distribution of the Gly16* allele, strains that harbor this allele are among the set of highly genetically divergent *C. elegans* strains (S6 Fig, S10 Table). However, not all of the genetically divergent strains harbor variation in *glct-3*.

To further explore the evolutionary history of the *glct-3* gene, we examined the conservation of *glct-3* paralogs and their orthologs. The *glct-3* gene encodes a glucuronosyl transferase-like protein that has six paralogs in *C. elegans*, including five closely related genes *glct-1*, *glct-2*, *glct-4*, *glct-5*, and *glct-6*, and one distantly related paralog *sqv-8*, which we will not discuss further (Fig 5A, S11 Table). Five *glct* genes (1–5) are located on an 80 kb region on chromosome I,

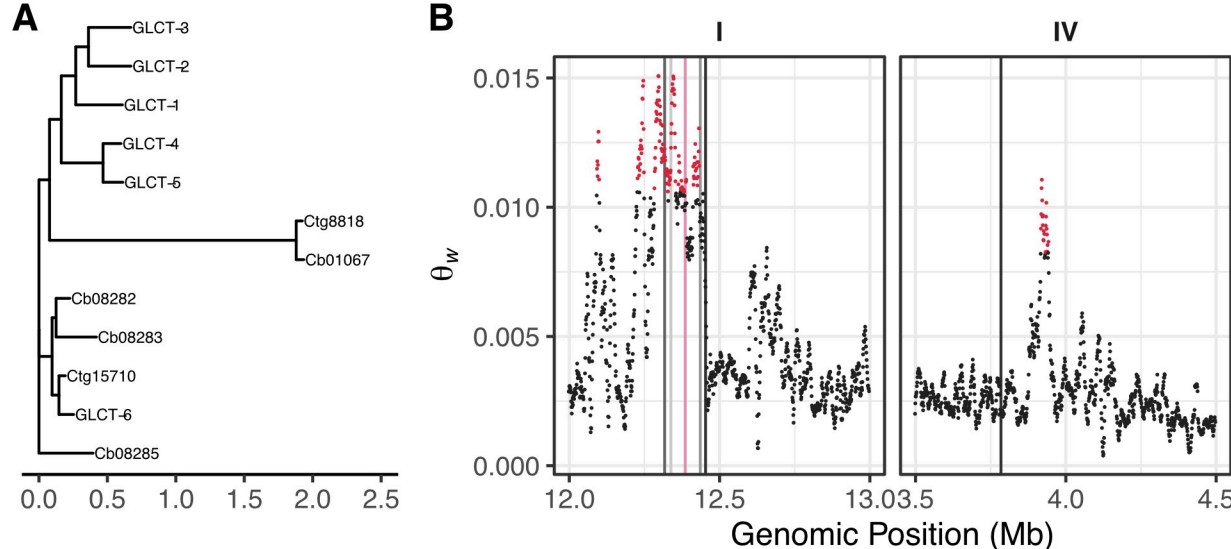

**Fig 5. Expansion of the *glct* family.** (A) An unrooted maximum-likelihood phylogeny of the six glucuronosyl transferase-like protein sequences encoded by the *C. elegans* genome and homologs from *C. briggsae* (Cb) and *C. tropicalis* (Ct). (B) The genetic diversity as measured by Watterson's theta (Θw) for the genomic regions that contain the six glucuronosyl transferase-like genes in *C. elegans* is shown. Each vertical line marks the position of a *glct* gene, from left to right on chromosome I: *glct-4*, *glct-1*, *glct-3* (pink), *glct-2*, and *glct-5*; and chromosome IV: *glct-6*. Each dot represents a 10,000 bp genomic region and is colored red if the Θw value is greater than the 99th quantile of values across the chromosome.

and *glct-6* is located on chromosome IV. The close proximity of five of the six paralogs suggests that these genes are the products of gene duplication events, as observed for other gene families in *C. elegans* [38–40]. We observed elevated levels of variation in the genomic region that contains *glct-1* through *glct-5* (**Fig 5B**, **S12 Table**), which supports the hypothesis that these sequences duplicated at some point in the *C. elegans* lineage and then diverged. Furthermore, the pattern of variation in the six *C. elegans glct* paralogs suggests that after the initial duplication event, the function of the *glct-6* gene was retained, which is indicated by the absence of deleterious variants in this gene among wild isolates (**Fig 5B**) [31]. Similarly, *glct-4* has no variation that is predicted to be deleterious. By contrast, *glct-1*, *glct-2*, *glct-3*, and *glct-5* contain variants predicted to have large effects on gene function. Among the 330 *C. elegans* strains, 24 have variation that is predicted to remove the function of two or more of these four genes, with two strains that have predicted loss-of-function alleles in all four genes.

## Copy number of the *glct* gene family varies across *Caenorhabditis* species

Next, we explored the conservation of the *glct* gene family across 20 species of *Caenorhabditis* nematodes, including ten for which the genome assembly was recently released [41]. We found that nine species contained only one *glct-3* ortholog, five contain two *glct-3* orthologs, three contain three *glct-3* orthologs, and one species each contain four, five, or six *glct-3* orthologs (**S7 Fig**, **S13 Table**). The prevalence of low-copy numbers of *glct* genes among a majority of *Caenorhabditis* species suggests that the ancestral copy number is fewer than the six copies found in the *C. elegans* genome. This hypothesis is supported by the presence of one and two *glct-3* orthologs in the outgroup species *Heterorhabditis bacteriophora* and *Oscheius tipulae*, respectively. Because the GLCT-6 protein and DNA sequences more closely resemble orthologous sequences among *Caenorhabditis* species than its paralogs in *C. elegans* (**S8 Fig**, **S14 Table**), this gene is likely the ancestral state of this gene family. Taken together, these results suggest that the copy number of *glct* genes likely affects fitness in the wild.

## Discussion

In this study, we identify mutations that naturally occur in *C. elegans glct-3* as modifiers of propionate sensitivity. This gene encodes a glucuronosyltransferase-like protein, which belongs to a family that includes the other GLCT proteins, as well as UGT enzymes. These enzymes generally catalyze the transfer of glucuronic acid to small molecules as part of the phase II detoxification system [42]. The addition of these adducts can make the small molecules more easily secreted or less able to interact with targets, decreasing the toxicity of these compounds. The mechanism by which mutations in *glct-3* render the animal less sensitive to exogenous propionate supplementation remains unclear. The closest human homologs of *glct-3* are B3GAT1, G3GAT2, and B3GAT3, but the functions of these genes are not well understood. Likewise, the family of UGT enzymes in *C. elegans* remain greatly understudied. These genes are likely involved in responses to the environment, so laboratory experiments often do not recapitulate the complex niches that *C. elegans* inhabit in the wild [8]. However, because natural variants in *glct-3* are predicted to cause loss of function, and the common Gly16* allele is an early nonsense allele, enzyme function is eliminated and GLCT-3 substrates cannot be modified and detoxified. This result suggests that GLCT-3 does not directly modify propionate or any of its derviaties as a detoxifying mechanism. Instead, our data indicate that modification of a small molecule, or perhaps protein, by GLCT-3 increases the toxicity of propionate. Future studies will determine which molecules are modified by GLCT-3 and what the functional consequences of such modifications are.

In the natural environment, *C. elegans* likely encounters a variety of bacteria and fungi that produce a plethora of small molecules, including short-chain fatty acids such as propionate. These small molecules can accumulate and decrease fitness in the niche. When bacteria that produce vitamin B12 are also present in the niche, propionate toxicity can be reduced. For these reasons, natural strains of *C. elegans* might vary in their complements of *glct-3* paralogs. Strains that inhabit niches with high propionate but low levels of vitamin B12 might have a more active propionate shunt [15] or fewer members of the *glct-3* family to limit toxicity. Niches with less propionate and/or high levels of vitamin B12 could support strains that might have lost the propionate shunt or have more copies *glct-3* family members. Microevolution of similar metabolic regulators could act through differences in copy number and not through specific changes to enzymatic function or differences in gene expression. Over the evolution of a particular pathway, individual components have the potential to expand and contract via duplication or deletion. As these changes occur, novel connections can compensate for the altered copy number of a particular pathway component. It appears that we have identified such a novel link in the *C. elegans* species, where a loss of *glct-3* function causes propionate resistance. It is still uncertain how our findings will translate to other systems, because it is possible that the phenomenon we observed is nematode-specific. It is clear, however, that natural changes in metabolic flux are important for how organisms deal with the complex milieu of their natural environment.

Interestingly, we could not validate the most significant QTL that we detected by GWA. This QTL on chromosome V was in strong LD with another QTL on chromosome V. Long-range LD is common in selfing organisms, especially *C. elegans* [19], but it can confound GWA mappings. Our results emphasize that all QTL need to be validated by independent strains. NILs, as we used here, offer an effective approach to rapidly test genomic intervals for correlations with observed phenotypic differences. It is likely that the chromosome V right QTL underlies the trait difference that we tested at the chromosome V left QTL. Additional NILs, obtained from crosses between different wild isolates, need to be constructed to test this hypothesis and narrow this genomic region to a causal gene.

Our study demonstrates that natural variation can modify sensitivity to the short chain fatty acid propionate. We used a *C. elegans* model that mimics metabolic conditions found in patients with propionic acidemia. These data indicate that *C. elegans* is a fruitful model to identify genetic modifiers of inborn errors in human metabolism, which is extremely difficult with human populations as these diseases are usually rare.

## Materials and methods

### Strains

All the wild strains were obtained from CeNDR (**S3 Table**) [31] and maintained at 20°C on nematode growth medium (NGM) plates on a diet of *E. coli* OP50. Near-isogenic lines (NILs) were generated using a procedure described previously [43] by crossing BRC20067 and DL238. Each NIL strain harbors recombination breakpoints at different locations on chromosome V generated by crossing two single recombinant strains, followed by six times backcrossing with BRC20067 to change the other five chromosomes into the BRC20067 background.

### Propionate sensitivity assays

A 2 M propionic acid stock solution was prepared in a chemical hood. For 40 ml solution, 6 ml propionic acid (sigma, #402907), 13.5 ml 5 M sodium hydroxide, and 20.5 ml water were mixed together, and the pH was adjusted to 6.0 with sodium hydroxide. The solution was filter sterilized and stored at 4°C. On day 0, arrested L1 animals were placed on seeded plates with

propionate and after incubation for two days, animals that developed beyond the L1 stage were evaluated as survivors. Propionic acid survival rate was calculated as the proportion of animals that have developed beyond the L1 stage over the total number of L1 animals at day 0. We focused on L1 survival because L1 larvae are more strongly affected by propionate than later developmental stages. For example, L1s that survive 100 mM propionate treatment will continue to grow to adulthood. However, high concentrations of propionate do not affect *C. elegans* hatching rate, which makes using this phenotype difficult for GWA mapping. Biological triplicate experiments with three technical replicates were performed. For the panel of 133 wild isolates, 100 mM propionate was used in five biological replicates, each with four technical replicates. A biological replicate is an independent growth of a strain. For each biological replicate, strains were chunked from a starved plate, grown for two generations, bleached, synchronized as L1s in M9, and plated out on 48-well agar plates with propionate. For each technical replicate, approximately 100 arrested L1s were transferred to an individual well of a 48-well plate. Because we had a high level of replication, we were not able to complete the entire GWA phenotyping in one experiment and had to phenotype the strains in three separate sets. For each set, we included six reference strains to verify that the batch conditions did not change significantly. To process the data, we first took the mean of the four technical replicates and removed biological replicate outliers, which were defined by 1.5 times the standard deviation from the mean. This procedure eliminated one of five biological replicates for 89 strains and two of five replicates for 11 strains. Next, we corrected the strain phenotype data for biological replicate and strain set using a linear model with the formula (phenotype ~ biological replicate + strain set). Finally, we took the mean of the residual phenotypes and performed association mapping (S3 Table).

To explore the effects of replicate and strain set, we performed an analysis of variance. To perform this analysis, we focused on the six strains that were included in all of the experiments. We found that the biological replicate (*p*-value = 0.026, Cohen's F = 0.119) and set (*p*-value = 0.012, Cohen's F = 0.134) had moderate but significant effects on strain phenotypes. Furthermore, we found no significant effect of technical replicate (*p*-value = 0.83, Cohen's F = 0.05), which is expected because technical replicates were drawn from the same strain preparation. As expected, these effects were all negligible after correction with the linear model described above.

## Heritability calculations

For dose-response experiments, broad-sense heritability ($H^2$) estimates were calculated using the *lmer* function in the lme4 package with the following linear mixed-model (phenotype ~ 1 + (1|strain)). $H^2$ was then calculated as the fraction of the total variance that can be explained by the random component (strain) of the mixed model. For the complete dose-response experiment, we calculated $H^2$ per dose. For the fine-scale dose response experiment, we subsampled three replicates twelve independent times for $H^2$ calculations.

## Power analysis

To determine the number of replicate measures we needed to collect for each wild isolate, we measured L1 survival of the DL238 strain after exposure to 100 mM propionic acid in 40 replicates. The 40 replicates consisted of eight technical replicates across five independent preparations of agar plates with propionate. For a range of mean differences (0.01 to 1, in increments of 0.01), we subsampled two to eight replicates for each of the five plate preparations 100 times and calculated the standard deviation of L1 survival for the subsamples. To calculate the power to detect a difference across a range of replicates and mean differences, we used the power.t.

test function in the pwr R package with the following parameters—n = number of subsampled replicates, delta = (0.01 to 1, in increments of 0.01), sd = mean of the standard deviation sub-samples, sig.level = 0.00001, alternative = "two.sided", type = "two.sample". With four technical replicates across five independent plate preparations, we were able to detect a 20% difference in means 80% of the time. We note that this analysis does not determine the power to detect QTL via GWA mapping, which will depend on the effect size of a given locus, the number of strains analyzed, the overall trait heritability, interactions among contributing loci, and other factors.

## Marker-based genome-wide association mappings

GWA mapping was performed using phenotype data from 133 *C. elegans* wild strains. We performed the same mapping procedure as described previously [43]. Briefly, genotype data were acquired from the latest variant call format (VCF) release (Release 20180527) from CeNDR that was imputed using IBDseq, with the following parameters: minalleles = 5%, r2window = 1500, ibdtrim = 0, r2max = 0.8 [44]. We used BCFtools to filter variants that had any missing genotype calls and variants that were below 5% minor allele frequency [45]. We used PLINK v1.9 to LD-prune the genotypes at a threshold of $r^2 < 0.8$, using—*indep-pairwise 50 10 0.8* [46], [47]). This genotype data set consisted of 59,241 markers that were used to generate the realized additive kinship matrix using the *A.mat* function in the *rrBLUP* R package [48]. These markers were also used for genome-wide mappings. However, because these markers still have substantial LD within this genotype set, we performed eigen decomposition of the correlation matrix of the genotype matrix using *eigs_sym* function in Rspectra package (https://github.com/yixuan/RSpectra). The correlation matrix was generated using the *cor* function in the correlateR R package (https://github.com/AEBilgrau/correlateR). We set any eigenvalue greater than one from this analysis to one and summed all of the resulting eigenvalues to obtain 772 independent tests within this genotype matrix [49]. We used the *GWAS* function in the rrBLUP package to perform genome-wide mapping with the following command: *rrBLUP:: GWAS(pheno = PC1, geno = Pruned_Markers, K = KINSHIP, min.MAF = 0.05, n.core = 1, P3D = FALSE, plot = FALSE)*. To perform fine-mapping, we defined genomic regions of interest from the genome-wide mapping as +/- 100 single-nucleotide variants (SNVs) from the rightmost and leftmost markers above the Bonferroni significance threshold. We then generated a QTL region of interest genotype matrix that was filtered as described above, with the one exception that we did not perform LD pruning. We used PLINK v1.9 to extract the LD between the markers used for fine mapping and the peak QTL marker identified from the genome-wide scan. We used the same command as above to perform fine mapping but used the reduced variant set. The workflow for performing GWA mapping can be found at https://github.com/AndersenLab/cegwas2-nf.

## Sequence Kernel Association Test (SKAT) Mapping

In parallel to marker-based GWA mappings, we performed a sequence kernel association test (SKAT), which is implemented in the RVtests software package using SKAT [34], [50].

We set the maximum allele frequency for SKAT to 50% using the—freqUpper from flag cegwas2-nf, and the minimum number of strains to share a variant to two using the—minburden flag. The model used for burden mapping was set with the following flag—kernel skat when executing rvtests. This flag implements the burden test that is represented in equation 1 of Wu *et al.* 2011 [34]. We used the weights suggested by Wu *et al.* 2011 when running the mapping (beta1 = 1, and beta2 = 25), which adds more weight to rare variants when testing a gene's variants for association with propionate responses.

## Linkage Disequilibrium

We used the *LD* function from the *genetics* package in R to calculate linkage disequilibrium and report the $r^2$ correlation coefficient between the markers (https://cran.r-project.org/package=genetics).

## Phylogenetic analysis

DNA and protein FASTA files for each species were downloaded from http://download.caenorhabditis.org/v1/sequence/ [41]. DNA and protein for each species FASTA files were combined and custom DNA and protein BLAST databases were built using *makeblastdb* [51]. The *glct-3* coding sequence (CDS) was used to query the DNA BLAST database using the *blastn* command with the *-evalue* threshold set to 1. Homologous sequences were extracted from the database using the *blastdbcmd* command. Next, a multiple sequence alignment of the homolgous sequences was generated using MUSCLE [52] with default settings and output in the phylip format.

For DNA sequences, the *raxmlHPC-AVX* command from RAxML 8 (v 8.2.12) with the GTRGAMMA substitution model was used to generate initial phylogenies [53]. Next, we preformed bootstrapping with the following command *raxmlHPC-AVX -p 12345 -x 12345 -# autoFC -m GTRGAMMA* and extracted the best tree with bootstrap support.

For protein sequences, we used the bayesian information criterion model selection feature of RAxML 8 to identify VT [54] as the best substitution model with the following command: *raxmlHPC-AVX -p 12345 -m PROTGAMMAAUTO—auto-prot = bic*. Next, we performed bootstrapping of the phylogentic tree using the following command: *raxmlHPC-AVX -p 12345 -x 12345 -# autoFC -m PROTGAMMAAUTO—auto-prot = bic*. All phylogenies were visualized using the interactive tree of life software [55] or the *ggtree* R package [56].

## Heritability estimates from genome-wide association phenotypes

We used the *sommer* package in R to calculate marker-based narrow-sense heritability [57]. We first calculated the marker-based heritability for the phenotype data used to perform GWA mapping using the *mmer* function in *sommer*, with the random variable in the model specified as *random = ~vs(strain, Gu = A) + vs(strainE,Gu = E)*, where A is the additive genotype matrix (generated using the *A.mat* function in *sommer*) and E is the epistatic genotype matrix (generated using the *E.mat* function in *sommer*). To determine the effect of the GLCT-3 GLY16* allele, we performed the same heritability calculation, but we included the presence of the GLCT-3 GLY16* allele as a fixed effect in the *mmer* function. Narrow-sense heritability estimates were extracted from the *mmer* object using the *sommer pin* function with the formula $h2 \sim (V1) / (V1+V2+V3)$, where V1 is the additive variance component, V2 is the epistatic variance component, and V3 is residual variation. To determine the effect of the GLCT GLY16* allele, we performed the following calculation *(h2—h2_glct3)/h2*, where *h2* is the narrow-sense heritability of the original phenotype data, and *h2_glct3* is the estimate with the GLCT GLY16* allele as a fixed effect.

## Supporting information

**S1 Fig. Power calculations.** Power analysis of L1 survival after propionate exposure is shown. We calculated power for a range of mean differences from 0 to 1, using the average standard deviation of 100 subsamples from a large-scale experiment that measured DL238 propionate survival. The solid line represents the mean of 10 replicate power calculations and the shaded area around the solid lines represent the standard deviation of the replicates. The line colors

represent the sample size. The dashed red line indicates 0.8 power.
(TIFF)

**S2 Fig. Experimental procedure and individual data sets.** (A) L1 survival in the presence of 100 mM propionate for six *C. elegans* strains with eight technical plus five biological replicates. (B) Experimental setup to phenotype wild isolates for GWA mapping. 133 wild *C. elegans* strains were divided into three batches to test their survival after exposure to 100 mM propionate. Each batch contains 48 strains, including six control strains that control for batch effects. (C) L1 survival rate in the presence of 100 mM propionate for each 48 strain batch described in B. The colored bars represent the mean L1 survival for each *C. elegans* strain. Tukey box-plots overlay the data. Batch-control strains are indicated by red median bars.
(TIFF)

**S3 Fig. Linkage disequilibrium of genomic loci significantly associated with propionate sensitivity.** Linkage disequilibrium ($r^2$) of peak QTL markers identified by genome-wide association mapping is shown. The tile color represents the correlation between marker pairs.
(TIFF)

**S4 Fig. Variation in chromosome I genes associated with propionate sensitivity.** (A) The normalized L1 survival in the presence of propionate for each phenotyped *C. elegans* strain is shown on the x-axis. The y-axis represents unique haplotypes (numbered from 1:n) constructed from variants with moderate-to-severe predicted effects on *glct-3* found to be significantly associated with propionate sensitivity. If a variant with a high predicted effect on gene function was identified, we plotted it separately. Therefore, a strain can be represented twice if it contains a variant with a high predicted effect on gene function. The red diamonds represent the median phenotype value for each unique haplotype. The blue and pink diamonds represent the DL238 and BRC20067 strains, respectively. (B) The pairwise linkage disequilibrium ($r^2$) between the allele that encodes the Gly16* (red diamond) in GLCT-3 and all variants (black diamonds) in the surrounding genomic region is shown on the y-axis. The x-axis represents the genomic position (Mb) of each variant.
(TIFF)

**S5 Fig. The global distribution of the GLCT-3 Gly16* allele.** Sampling locations of wild *C. elegans* strains. Each dot represents the location where an individual strain was sampled. Pink dots represent strains carrying the REF allele at GLCT-3, and blue dots represent strains carrying the Gly16* allele.
(TIFF)

**S6 Fig. Phylogenetic tree of the C. elegans population.** A maximum likelihood phylogenetic tree of the *C. elegans* population. Branches are colored based on the GLCT-3 allele the individual strain carries, blue represents strains with the GLCT-3 Gly16* allele, and pink represents strains with the reference allele.
(TIFF)

**S7 Fig. Phylogenetic relationship of glct-3 homologous cDNA sequences.** The maximum likelihood phylogenetic relationship of *glct-3* homologs is shown. Branch lengths are shown above each branch. Branch colors correspond to the bootstrap support for the split, with pink indicating higher support. If a species contains more than homolog, all homologs for that species are colored the same color. Species with only one homolog are colored black. The *C. elegans glct* genes are colored in black and bolded.
(TIFF)

**S8 Fig. Phylogenetic relationship of GLCT-3 homologous protein sequences.** The maximum likelihood phylogenetic relationship of *glct-3* homologs is shown. Branch colors correspond to the bootstrap support for the split, with pink indicating higher support. The *C. elegans* GLCT protein sequences are bolded.
(TIFF)

**S1 Table. Raw propionate dose response (0–140 mM) phenotypes.**
(TSV)

**S2 Table. Raw propionate dose response (80–110 mM) phenotypes.**
(TSV)

**S3 Table. Processed L1 survival phenotypes used for GWA mapping.**
(TSV)

**S4 Table. Genotypes used for GWA mapping.**
(TSV)

**S5 Table. Results from single-marker mapping.**
(TSV)

**S6 Table. Results from SKAT mapping.**
(ASSOC)

**S7 Table. Chromosome V NIL genotypes.**
(TSV)

**S8 Table. Raw L1 phenotypes from near-isogenic lines.**
(TSV)

**S9 Table. Raw L1 phenotypes from genome-edited strains.**
(TSV)

**S10 Table. *C. elegans* genome-wide relatedness.**
(TXT)

**S11 Table. GLCT homolog phylogeny.**
(TXT)

**S12 Table. Watterson's theta for chromosomes I and IV.**
(TSV)

**S13 Table. DNA phylogeny of *glct* homologs.**
(TXT)

**S14 Table. Protein phylogeny for GLCT homologs.**
(TXT)

## Acknowledgments

We thank members of the Andersen and Walhout laboratories for critical comments on the manuscript.

## Author Contributions

**Conceptualization:** Huimin Na, Stefan Zdraljevic, Albertha J. M. Walhout, Erik C. Andersen.

**Data curation:** Huimin Na, Stefan Zdraljevic, Albertha J. M. Walhout, Erik C. Andersen.

**Formal analysis:** Huimin Na, Stefan Zdraljevic, Robyn E. Tanny, Albertha J. M. Walhout, Erik C. Andersen.

**Funding acquisition:** Albertha J. M. Walhout, Erik C. Andersen.

**Investigation:** Huimin Na, Stefan Zdraljevic, Albertha J. M. Walhout, Erik C. Andersen.

**Methodology:** Huimin Na, Stefan Zdraljevic, Robyn E. Tanny, Albertha J. M. Walhout, Erik C. Andersen.

**Project administration:** Albertha J. M. Walhout, Erik C. Andersen.

**Supervision:** Albertha J. M. Walhout, Erik C. Andersen.

**Validation:** Huimin Na, Stefan Zdraljevic, Albertha J. M. Walhout, Erik C. Andersen.

**Visualization:** Huimin Na, Stefan Zdraljevic, Albertha J. M. Walhout, Erik C. Andersen.

**Writing – original draft:** Huimin Na, Stefan Zdraljevic, Albertha J. M. Walhout, Erik C. Andersen.

**Writing – review & editing:** Huimin Na, Stefan Zdraljevic, Robyn E. Tanny, Albertha J. M. Walhout, Erik C. Andersen.

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
