## [Decision Letter · Decision Letter 0]

2 Apr 2020

Dear Erik,

Thanks for your patience on waiting for this decision.

Thank you very much for submitting your Research Article entitled 'Natural variation in a glucuronosyltransferase modulates propionate sensitivity in a Caenorhabditis elegans propionic acidemia model' to PLOS Genetics. Your manuscript was fully evaluated at the editorial level and by independent peer reviewers. The reviewers appreciated the attention to an important problem, but raised some substantial concerns about the current manuscript. Based on the reviews, we will not be able to accept this version of the manuscript, but we would be willing to review again a much-revised version. We cannot, of course, promise publication at that time.

This work is thoughtful and a useful demonstration of model system genetics and use of the CeNDR platform to conduct GWA inquiries. The reviewers in particular note that inclusion of the chrom V QTL is valuable, even though it turned out to be a false positive.

The main concern is about the larger biological context for this work. Reviewer 3 states that the major motivation for this paper is to understand human disease, but there is not much discussed on this point, nor on other studies seeking to identify modifier variants. Reviewer 1 is concerned about the interest/impact level of this work to the PLOS Genetics audience, and specifically mentions a low level of discussion around possible GLCT mechanism. Like these reviewers, I also feel that the results in this manuscript don't quite meet the stated goal of the project--to use the tractable C. elegans model to identify variants/genes affecting propionate sensitivity, and thus gain insight into modifiers of human disease. (It would appear, based on references in the Intro, that the presence of modifier/genetic background effects for inborn errors in human metablic disease is already well established. If so, the fact that this study illustrates functional variation in proprionate sensitivity in natural populations is not especially impactful; but if that is not well established, perhaps an expanded discussion of the outstanding question(s) about variation in the human disease context would be valuable.) Given the idiosyncrasy of the C. elegans model, including GLCT gene family expansion, connecting even mechanistic findings in worms (which this work has not yet reached) to potential impact in human disease could still be hard, and still suffer from the same concern about relevance, so I fully acknowledge the challenge here. But, as written, it feels as though the potential biological insights that may be gleaned from this work have not been fully considered. Can the authors expand (more) upon the possible meaning behind showing that a putative glucuronosyltransferase affects proprionate sensitivity? Only one paper is cited in the Discussion describing the function of glucuronosyltransferases. Is very little known about these enzymes?

Reviewers 2 and 3 both requested expanded explanations of methods, including better descriptions of the replication scheme and some specific requests for illustrating or reporting variation in the data. I also think Reviewer 2's points 6) and 7) are good ones and the ms will be well served by clarifying them. In addition to their comments below, I also was curious about how "GLCT function," generally interpreted, appears to associate with proprionate sensitivity. Of the six genes, GLCT-4 and GLCT-6 don't have much variation, but the other four do. Obviously strain DL238 has the stop-gain in GLCT-3 and the authors report that multiple strains have putative loss of function at two, three or four genes. What is DL238's status at the other GLCT genes? Of the wild isolates phenotyped, how many of them had putative loss of function at variable numbers of genes? Enough to ask (perhaps qualitatively) how loss of function across the GLCT family associates with proprionate sensitivity? Perhaps this was already explored, but I didn't see it mentioned in the ms. I'm curious about this because in my mind it goes to the biological context question raised above. If a more pointed hypothesis can be made about glucuronosyltransferase activity level and proprionate sensitivity, then the results speak to a potentially new biological insight, rather than a worm-specific, potentially even GLCT-3-specific, mechanism.

If you decide to revise the manuscript for further consideration at PLOS Genetics, please aim to resubmit within the next 60 days, unless it will take extra time to address the concerns of the reviewers, in which case we would appreciate an expected resubmission date by email to plosgenetics@plos.org.

[LINK]

We are sorry that we cannot be more positive about your manuscript at this stage. Please do not hesitate to contact us if you have any concerns or questions.

Yours sincerely,

Annalise Paaby

Guest Editor

PLOS Genetics

Gregory P. Copenhaver

Editor-in-Chief

PLOS Genetics

Reviewer's Responses to Questions

**Comments to the Authors:**

Reviewer #1: In this study, the authors used genome-wide association studies in C. elegans to identify genetic regulators of propionic acidemia. They identified one regulator, a gene encoding a putative glucuronosyltransferase. They generated a knockout allele of the gene, which confirmed its role in propionate toxicity. This work was carefully carried out, and a role for GLCT (B3GAT) in propionate metabolism is interesting, but puzzling.

At this point, with no mechanistic data or barely any speculation about how GLCT could be acting to render worms sensitive to propionate, it is hard to get excited about these results, and it does not seem like this work will generate general interest among readers.

Reviewer #2: Dear authors,

This paper presents a thorough set of experiments that lead to the identification of natural allelic variation in glct-3 linked to propionic acidemia in the model nematode Caenorhabditis elegans. The authors follow a clear experimental line: screening a limited set of wild isolates, screen a set of 133 wild isolates, follow-up by creation of introgression lines and a target-gene approach, after which the authors explore natural variation within and beyond the species for the glct gene family.

There are various interesting aspects in this study, including an interesting approach to the GWAS analysis (combining ‘classical’ GWA with burden-testing) and two-types of follow-up experiments. Here, I think, the authors should be applauded for including the results from the introgression line experiments, even though they did not lead to the identification of a gene or mechanism. Of course, I do have some comments and requests for clarification

Major comments:

1. The authors do their best to explain how they conducted the experiments in replicates and how they dealt with the effects of having to split up the experiments in batches. However, it is not described what exactly constitutes a ‘technical’ and a ‘biological’ replicate, nor is it clear what exactly represents a ‘batch’. As the authors amass a large dataset on these factors, I would like the paper to include the actual effect of these levels of replications. These could be included in the materials and methods before the batch correction. An analysis of variance on these factors before and after batch correction would be good to assess the impact of this correction on the data.

2. Next to presentation in the figures, can a table be included of both GWA analyses with the location of all variants above the threshold, the alt allele, the effect, and the variance explained?

3. In the discussion, in the last paragraph, the authors remark ‘that C. elegans is a fruitful “simple” model to identify genetic modifiers of inborn error in human metabolism’. While I fully agree with the idea that in C. elegans genetic and environmental control allow identification of pathway components that could easily have eluded human geneticists, the results presented here on the expansion of the glct gene family present another side of the coin. I would like to read the authors opinion on how a gene family expansion affects what we can infer from C. elegans as a model for human metabolic diseases. Is the canonical strain N2, as referred to in the introduction, the best genetic basis for studying propionic acidemia?

Minor comments (some have overlap with the major comments):

Line 84: ... toxicity, likely ...

Line 86-87: ‘The vast ... Bristol, England’. The point made here is very implicit and it would be good for readers to connect it more explicitly to the next sentence

Line 129-131: I think I’m clear on what ‘biological replicate’ means, however what constitutes a technical replicate is not described (neither in the M&M).

Line 137: It seems this should be figure 1D?

Line 144-146: ‘To identify ... (H2= 0.79)’ Can you explain the rationale of subsampling for the detailed concentration range and not for the extended range?

Line 154 – 156: ‘We tested ... effects (Figure S2)’. I got a bit confused here, is it correct the setup is that all 5 biological replicates with 4 technical replicates were done within the same batch? Together with my previous question on what constitutes a technical replicate this does not really clear things up. I currently read it as: we measured 20 ‘replicates’ of a strain at the same time and anchored these measurements with 6 strains included along all these measurements.

Line 168: ‘Therefore, we ... gene basis [34]’. How many tests were conducted by SKAT; how many genes passed the thresholds set?

Line 215: ‘this gene is a major contributor’

Line 222: why is Cohen’s F reported here, whereas in the figure a t-test is used?

Line 222-226: ‘The strains ... to propionate’. How does that compare to the effect size of the QTL detected?

Line 340-342: ‘To process ... standard deviation’ how many datapoints were removed?

Line 345: ‘as above’; please describe how exactly, it is not clear of what you remove the outliers? How many values were removed?

Figure 1

General: how many larvae were tested?

Line 654-660: It is really hard for me to see lines in that figure, maybe they are completely over plotted, but then mention that.

Line 663: was this corrected for multiple testing?

Line 664-667: initially looking at the figures and the legend, this confused me, can you mention it is another experiment.

Line 668: should be ‘shown in B’)

Figure 5

General (B): what do the vertical lines indicate? Which of the glct genes?

Line 722: Please state what Watterson’s theta indicates.

Figure S4

General (A): can a strain be represented multiple times? It seems that having a stop, for example, often co-insides with Hap:3?

Reviewer #3: The authors describe a study to map propionate sensitivity using the C. elegans model system. They take a GWAS approach using a set of 133 wild strains and identified several QTL influencing this trait. For one QTL, they validate a candidate gene, while for a second QTL, validation failed. Overall, this is nice study showing how an animal model can help gain insights into human diseases that are difficult to study. I appreciated the open and thorough discussion of the negative results for the QTL on chromosome V. I do have some concerns and comments, that I detail below and hope will improve the manuscript.

1) There are several places where the experimental procedures or data analysis is not explained fully. For example, it would be useful to know the number of worms assayed on each plate and to have an explanation of what constitutes a technical versus biological replicate in this experiment. Maybe I am missing something but I am not sure what a “technical” replicate is for a study quantifying survival of a group of individuals. In addition, I think it would be appropriate to briefly describe the statistical model that is fit by the SKAT approach. There are many different types of gene-based association approaches and it would be helpful to have a bit more information about the approach they have taken. Another example is in Figure 2B, where they refer to the Eigen-decomposition significance threshold but don’t explain this threshold in the Methods.

2) The assay focused on L1 stage survival. How well does this model the human disease? Did the authors consider using survival to (or at) another stage? Some justification of this experimental decision would be helpful.

3) The presented data are all in proportions. Was the data analyzed as proportions or was a statistical model using a binomial distribution used? This wasn’t clear to me in the Methods. If the raw proportions were used, is the starting number always the same (meaning there is the same amount of uncertainty for all estimates?) In some cases, the median and mean look to be pretty different. What does the distribution of the data and the residuals look like? I would like to see a more in depth discussion of these issues and a justification for the appropriateness of the statistical model for their data.

4) Determination of significance: The authors chose to use a Bonferroni threshold. I would advocate for a false discovery approach instead since Bonferroni is already known to be overly conservative and they are not highly powered with just 133 strains so the risk of type II errors is high.

5) The power analysis indicates the number of replicates needed to detect a phenotypic pairwise difference between 2 strains (via a single t-test). This analysis does not indicate the power of the GWAS, which will depend on the effect size of the locus, the heritability of the trait overall (how many other loci contribute), and the number of strains, etc. To avoid any confusion for the reader, I think the authors should state this very clearly so the impression is not given that the analysis indicates the power of the GWAS.

6) Patterns of LD: Given the LD patterns observed in these strains, I expected the confidence intervals for the QTLs to be determined by how far strong LD extends, rather than a constant SNP number (100 SNPs). Perhaps the authors could provide a justification for this CI? For the gene-based test, I also think it might be valuable to consider a confidence interval based on LD. From Figure 4a, it looks like LD at least has the potential to drive false signals.

7) I commend the authors for including a frank discussion of the lack of validation for the QTL on chromosome V. Have the authors considered whether the strains chosen for the validation might be driving the lack of result? Given this is a quantitative trait, there are presumably many variants in the genome that influence this trait and the uncertainty associated with localizing a single causative marker, perhaps the two strains chosen did not harbor the causative allele at this QTL.

8) The major motivation for the paper is presented as a way to understand human disease but there is not too much discussion of this point or much background given for other studies focused on identifying modifier loci. The authors might consider expanding these points. Or, if not, they might consider whether a combined Results & Discussion section might serve the paper better.

9) Figures:

In Figure 1B, I think there should be 12 lines based on what I think is being plotted but I only see maybe 6.

In Figure 1E, it would be useful to also plot estimated standard errors. Often for heritability estimates, these can be obtained using a jackknife approach. There is also an error in the legend for 1E – I believe they intended to refer to 1B not 1A.

In Figure 2A, if it is possible to plot the raw data points with means and SEs overlaid, I always prefer that to a dynamite plot. The color choices in B are much to close. A lighter pink paired with a darker red might work but 2 more divergent colors would be better.

**Have all data underlying the figures and results presented in the manuscript been provided?**

Reviewer #1: Yes

Reviewer #2: Yes

Reviewer #3: Yes

PLOS authors have the option to publish the peer review history of their article (what does this mean?). If published, this will include your full peer review and any attached files.

Reviewer #1: No

Reviewer #2: Yes: Mark G. Sterken

Reviewer #3: No

---

## [Decision Letter · Decision Letter 1]

2 Jul 2020

Dear Erik and Marian,

Thank you for your thoughtful consideration of the initial reviews and also for your patience in waiting for the decision. All three of the original reviewers have read your revised manuscript, as have I, and we are in agreement that the revisions are substantial improvements, addressing all previous concerns and positioning this work as a valuable contribution to PLOS Genetics.

Reviewer 2 has raised several additional minor points, so I offer you the opportunity to address these in a final revision. Please note that his third point (about the amount of research currently addressing ugt-like genes and whether/how that invites speculation about their function in propionate sensitivity) is a critical one, as it goes to the heart of the original concern with the manuscript, which was about placing your results in the context of human health or metabolic mechanisms generally. However, since your revisions on this point were in response to my asking about the extent of the literature on these genes, and also because Reviewer 2 is not raising this point to challenge the overall impact of the work (he was enthusiastic about the manuscript's interest level from the beginning), I leave it up to you how to address it.

Best,

Annalise

Thank you very much for submitting your Research Article entitled 'Natural variation in a glucuronosyltransferase modulates propionate sensitivity in a Caenorhabditis elegans propionic acidemia model' to PLOS Genetics. Your manuscript was fully evaluated at the editorial level and by independent peer reviewers. The reviewers appreciated the attention to an important topic but identified some aspects of the manuscript that should be improved.

We therefore ask you to modify the manuscript according to the review recommendations before we can consider your manuscript for acceptance. Your revisions should address the specific points made by each reviewer.

[LINK]

Yours sincerely,

Annalise Paaby

Guest Editor

PLOS Genetics

Gregory P. Copenhaver

Editor-in-Chief

PLOS Genetics

Reviewer's Responses to Questions

**Comments to the Authors:**

Reviewer #1: I am satisfied with the changes the authors made in the text. The relevance of their worked is now explained in a more clear manner.

Reviewer #2: Dear authors,

Thank you for addressing my concerns and questions, the paper has been much improved and clarified. The results presented are interesting as they provide the research community with the evidence of a functional mutation by genome editing and a phylogenetic analysis across the Caenorhabditis genus. Combined these provide a sufficient step and good grounding for starting subsequent research by these two groups or others aimed at a clarification of the exact mechanism.

I do have three remarks:

Line 167: why is the heritability 0.53? From the figure I read something like 0.7 (also compared to the value in the previous version this is low).

Line 260: the narrow-sense heritability is mentioned, but I cannot find the methods of how this was estimated.

Line 320-326: I was not particularly helped by these statements in the discussion. Why do 50 references in human literature and a dozen references in C. elegans literature not provide enough ground for speculation? If current literature is not sufficient it would be more helpful to either discuss where the gaps are or options that are on/off the table. In other words, can you specify what makes it difficult to speculate? Because ‘only’ 50 or a dozen hits are not the reason this speculation is difficult.

Reviewer #3: The authors have done a thorough job revising their manuscript and responding to the comments from the reviewers. I particularly appreciated the expansion of the Methods section to clarify several points. Overall, I feel my concerns have been addressed by this revision.

**Have all data underlying the figures and results presented in the manuscript been provided?**

Reviewer #1: Yes

Reviewer #2: Yes

Reviewer #3: Yes

PLOS authors have the option to publish the peer review history of their article (what does this mean?). If published, this will include your full peer review and any attached files.

Reviewer #1: No

Reviewer #2: **Yes: **Mark G. Sterken

Reviewer #3: No

---

## [Editor Report · Decision Letter 2]

8 Jul 2020

Dear Erik,

We are pleased to inform you that your manuscript entitled "Natural variation in a glucuronosyltransferase modulates propionate sensitivity in a Caenorhabditis elegans propionic acidemia model" has been editorially accepted for publication in PLOS Genetics. Congratulations!

Yours sincerely,

Annalise Paaby

Guest Editor

PLOS Genetics

Gregory P. Copenhaver

Editor-in-Chief

PLOS Genetics

Comments from the reviewers (if applicable):

**Data Deposition**

http://datadryad.org/submit?journalID=pgenetics&manu=PGENETICS-D-20-00143R2

**Press Queries**

---

## [Editor Report · Acceptance letter]

17 Aug 2020

PGENETICS-D-20-00143R2 

Natural variation in a glucuronosyltransferase modulates propionate sensitivity in a Caenorhabditis elegans propionic acidemia model 

Dear Dr Andersen, 

We are pleased to inform you that your manuscript entitled "Natural variation in a glucuronosyltransferase modulates propionate sensitivity in a Caenorhabditis elegans propionic acidemia model" has been formally accepted for publication in PLOS Genetics! Your manuscript is now with our production department and you will be notified of the publication date in due course.

With kind regards,

Kaitlin Butler

PLOS Genetics

On behalf of:
